# Characteristics of Mucosa-Associated Microbiota in Ulcerative Colitis Patients with 5-Aminosalicylic Acid Intolerance

**DOI:** 10.3390/biomedicines12092125

**Published:** 2024-09-19

**Authors:** Hiroshi Matsumoto, Momoyo Sasahira, Tei Tei Go, Shogen Yo, Takehiro Ninomiya, Motoyasu Osawa, Osamu Handa, Eiji Umegami, Ryo Inoue, Akiko Shiotani

**Affiliations:** 1Department of Gastroenterology, Kawasaki Medical School, Okayama 701-0192, Japan; momomo0318@gmail.com (M.S.); gutingting0529@med.kawasaki-m.ac.jp (T.T.G.); yoshogen@med.kawasaki-m.ac.jp (S.Y.); takehiro.nino0614@gmail.com (T.N.); o.m.1976-1017@med.kawasaki-m.ac.jp (M.O.); handao@med.kawasaki-m.ac.jp (O.H.); eumegaki@med.kawasaki-m.ac.jp (E.U.); shiotani@med.kawasaki-m.ac.jp (A.S.); 2Faculty of Agriculture, Setsunan University, 45-1 Nagaotoge-cho, Hirakata 573-0101, Japan; ryo.inoue@setsunan.ac.jp

**Keywords:** gut microbiota, mucosa-associated microbiota, ulcerative colitis, 5-ASA intolerance, dysbiosis

## Abstract

Background/Objectives: 5-Aminosalicylic acid (5-ASA) is a first-line therapy for ulcerative colitis (UC). This study examined the mucosa-associated microbiota (MAM) in UC patients, distinguishing between those who were 5-ASA tolerant and intolerant. Methods: Brushing samples were collected from the sigmoid and ileal end of patients with UC during endoscopic procedures. The samples were profiled by using the Illumina MiSeq platform. The V3–V4 regions of the 16S rRNA gene (460 bp) were amplified by using tailed PCR. Results: A total of 15 patients with 5-ASA intolerance, 38 patients with 5-ASA tolerance, and 19 healthy controls were recruited in this study. The α-diversity indices were remarkably different among the three groups in the ileum mucosa but not in the sigmoid colon. In the ileum mucosa, Alistipes, Ruminococcaceae, and Odoribacter were less abundant in the 5-ASA-intolerant group than in the control and 5-ASA-tolerant groups. On the contrary, Merdibacter, Brevundimonas, and Porphyromonas were more abundant in the 5-ASA-intolerant group than in other groups. Conclusions: The present study showed that the changes in MAM were characterized by a decrease in mucoprotective bacteria rather than an increase in harmful bacteria.

## 1. Introduction

Ulcerative colitis (UC) is a chronic inflammatory disease that impacts the colon and rectum [1]. In genetically susceptible individuals, intestinal inflammation occurs due to an overactive immune response and imbalanced intestinal microbiota. However, the mechanism of UC remains unclear [2,3,4]. Bacterial dysbiosis in IBD is characterized by a reduction in bacterial diversity, a decrease in the Firmicutes phylum (Faecalibacterium, Blautia, Roseburia, et al.), and an increase in the Proteobacteria phylum (Enterobacteriaceae, including Eschrichia) [5,6,7]. Compared with healthy subjects, an increase in fecal Proteobacteria and a decrease in Firmicutes have been observed in IBD patients; additionally, compared with patients in remission, the mucosa of patients in an active stage were colonized with a higher abundance of Proteobacteria and a lower abundance of Firmicutes.

Treatments for UC are based on the severity of the symptoms and may include corticosteroids, biological therapies, small molecular drugs, adhesion molecule inhibitors, azathioprine or 6-mercaptopurine, and 5-aminosalicylic acid (5-ASA). For patients with mild-moderate UC, 5-ASA is the conventional treatment for the induction of remission. A 5-ASA-containing preparation is the first-line option for induction and maintenance therapy in patients with mild-to-moderate UC. 5-ASA-containing preparations have two major types, namely, mesalazine/5-ASA and sulfasalazine (SASP), which are currently used as first-line therapy for UC [8,9,10]. SASP is composed of 5-ASA linked to sulfapyridine by a diazo bond, and this bond is readily cleaved by bacterial azoreductases in the colon [11]. 5-ASA is absorbed by intestinal epithelial cells, providing anti-inflammatory benefits to the intestinal mucosa. 5-ASA may neutralize oxidative stress inflicted by macrophages and neutrophils, preventing leukotriene synthesis and the recruitment of inflammatory cells [12]. Mehta R et al. reported that the gut microbial metabolism of 5-ASA diminishes its clinical efficacy in inflammatory bowel disease (IBD) [13]. Twelve uncharacterized microbial acetyltransferases, which belong to thiolases and acyl-CoA N-acyltransferase superfamilies, are linked to 5-ASA inactivation.

Recent experimental data suggest that the gut microbiome plays a role in the pharmacokinetics of several medications [14,15,16]. For example, gut microbial enzymes were recently identified and characterized in vitro to metabolize digoxin and L-dopa, medications for heart failure and Parkinson’s disease, respectively. In humans, association studies suggest a role of the microbiome in modulating the drug efficacy of anti-cytokine biologics and cardiometabolic drugs [17,18]. In addition, it focused on reports that the intestinal microbiota plays a significant role in the efficacy of immune checkpoint inhibitors, anti-programmed cell death protein 1 receptor (PD-1) therapy and cytotoxic T-lymphocyte-antigen (CTLA-4), against cancer [19,20]. Although they generally lack mechanistic explanations, drug metabolism by gut microbiota is associated with the effects and adverse events of 5-ASA and other drugs and may also be related to the clinical outcomes of the disease stage.

5-ASA and SASP are effective and safe, but some patients may become intolerant to 5-ASA [21,22]. 5-ASA can cause intolerance in up to 15% of patients, despite having few serious side effects [23,24,25]. Headache, skin rash, nausea, abdominal pain, and diarrhea comprise the list of mild adverse events. More severe adverse events include nephrotoxicity, hepatic dysfunction, pancreatitis, pericarditis, pneumonitis, severe skin reaction, toxic epidermal necrolysis, and acute gastroenteropathy, which is often referred to as “acute 5-ASA intolerance syndrome [26]”. 5-ASA intolerance mechanisms include allergic reactions, chemical toxicity, and dysbiosis [27,28]. Mizuno S et al. reported that the 5-ASA-intolerant group had more fecal bacteria in the genera *Faecalibacterium*, *Streptococcus*, and *Clostridium* than the 5-ASA-tolerant group [29]. However, there are only reports about the association between 5-ASA intolerance and mucosa-associated microbiota (MAM).

Most studies on the gut microbiome have focused on fecal microbiota, which is predominantly found in the lumen. Meanwhile, stable MAM has received increased attention [30,31]. The precise and accurate evaluation of MAM is crucial for comprehending and distinguishing host–microbe interactions contributing to health promotion and diseases. In previous studies, we found differences in bacterial community profiles between endoscopic brushing samples and fecal samples. During colonoscopic procedures, brushing samples may replace the need for fecal samples in analyzing MAM [32].

The relationship between UC and MAM has already been reported [33]. The reduction in the abundance of butyric acid-producing bacteria, *Odoribacter*, in ileal MAM may play an important role in the pathophysiology of UC. In this study, the characteristics of MAM of 5-ASA intolerance in patients with UC were investigated. The relationship between 5-ASA intolerance and MAM was also examined to devise a therapeutic strategy for addressing 5-ASA intolerance. This study aims to characterize MAM in UC patients with 5-ASA-intolerant patients.

## 2. Materials and Methods

### 2.1. Ethics

Ethical approval was obtained from Kawasaki Medical School Ethic and Medical Research Committee (no.3087; Ethics Committee approval date is 18 May 2020, specific dates for study period 27 June 2020–21 June 2024). Written informed consent was achieved from each research subject before enrollment. All patients were enrolled at the Division of Gastroenterology of Kawasaki Medical School Hospital. This study was registered at the University Hospital Medical Information Network Center.

### 2.2. Patients and Sample Collection

The enrolled subjects were as follows: healthy controls (HCs) undergoing routine medical checkups and patients with UC. The diagnosis of UC is based on a combination of clinical, biochemical, stool, endoscopic, and histological investigations [34]. As HCs, those who received any medication or complained of any gastrointestinal symptoms and patients with tumor lesions or inflammatory lesions found by colonoscopy were excluded. Only patients without antibiotic use within 3 months, colon cancer, history of colectomy, active GI bleeding, or hospital admission in the preceding 6 months were included. Samples were gathered from the terminal ileum and sigmoid colon using an endoscopic microbiology brush (COOK, Bloomington, IN, USA), following the standard polyethylene glycol preparation for colonoscopy. We analyzed the ileal mucus because we have previously reported that the ileal MAM is associated with severity and disease type in patients with UC [33]. Disease severity was evaluated on the basis of the Mayo endoscopic sub-score consisting of a four-point scoring system. 

### 2.3. Definition of 5-ASA Intolerance

As defined in previous reports [23,24,35], 5-ASA intolerance is diagnosed when individuals have exhibited at least one symptom—headache, digestive symptoms, skin symptoms, or fever—upon 5-ASA administration.

### 2.4. DNA Extraction, PCR Amplification, Library Preparation, and 16SrRNA Sequencing

16SrRNA gene amplicon library was prepared and sequenced using previously reported methods for DNA extraction, amplicon generation, and sequencing [32]. A library of amplicons encoding the 16S rRNA gene was prepared, and sequencing was conducted as previously described [36]. The samples were profiled by high-throughput amplicon sequencing with dual-index barcoding using the Illumina Miseq platform (Illumina, San Diego, CA, USA). 16S rDNA’s V3–V4 regions, spanning 460 base pairs, were amplified using tailed PCR [37]. Afterward, PCR amplicons were purified using SPRI select beads (Beckman Coulter, Brea, CA, USA). The DNA concentration of purified amplicons was measured using a Quantus Fluorometer and the QuantiFluor^®^ dsDNA System (Promega, Madison, WI, USA), and an approximately equal amount of their DNA was pooled. The pooled sample was sequenced using the Miseq Reagent Kit V3 (600 cycles; Illumina, San Diego, CA, USA) on the Miseq system in accordance with the manufacturer’s instructions.

### 2.5. Bioinformatics Analysis

Sequence data processing, consisting of quality filtering, chimera checking, operational taxonomic unit (OTU) definition, and taxonomic assignment, was accomplished using the combination of QIIME 1.9.0 (https://qiime2.org/), USEARCH 9.2.4 (https://cryptick-lab.github.io/NGS-Analysis/_site/usearch-previous.html), UCHIME 4.2.40 (https://www.drive5.com/usearch/), and VSEARCH 2.4.3 (https://github.com/torognes/vsearch/releases/tag/v2.28.1). Singletons were eliminated, and 97% sequence similarity OTUs were taxonomically assigned using the RDP classifier v2.10.2 with the Greengenes database version 13.8. 

### 2.6. Alpha-Diversity

The observed features, Shannon entropy, and Chao 1 diversity indices were calculated by using the phyloseq package of R software (version 4.4.1). It was visualized by principal coordinate analysis and statistically analyzed using permutational multivariate analysis of variance (PERMANOVA) using R software. PERMANOVA was performed by using the vegan package (version 2.6-2).

### 2.7. Statistical Analysis

The differences in families among the lines were analyzed using the Statistical Analysis of Metagenomic Profiles (STAMP) software (version 2.1.3) [38]. The diversity and relative abundance of bacterial genera indices were compared among three groups by the Kruskal–Wallis analysis and compared between the two groups by the Mann–Whitney U test. The category data were analyzed using chi-square test. Statistical analyses were performed using SPSS (version 25 for Windows, IBM Japan, Ltd., Tokyo, Japan). Statistical significance was set at a *p* value of <0.05. 

## 3. Results

### 3.1. Patient Background

Samples were obtained from 15 patients with UC who exhibited intolerance to 5-ASA (six men, mean age 48.1 years old), 38 patients with UC who demonstrated tolerance to 5-ASA (16 men, mean age 44.5 years old), and 19 controls (seven men, mean age 56.2 years old). The mean age was significantly older in the controls than in the UC group (Table 1). No significant difference in disease duration, disease type, type of 5-ASA, and endoscopic activity was found between the 5-ASA-tolerant and -intolerant groups. However, significant differences in biologics use were found; the 5-ASA-intolerant group used biologics more than the 5-ASA-tolerant group. In the 5-ASA-intolerant group, six patients showed negative results in the drug-induced lymphocyte stimulation test.

### 3.2. Analysis of Mucosa Samples

#### 3.2.1. The α-Diversity Indices among the Three Groups Using Ileum Mucus

The median indices of observed features and chao1 were significantly lower (*p* < 0.01) in the 5-ASA-intolerant group than in the controls. Moreover, significant differences in the observed features, Shannon entropy, and Chao1 indices were found among the three groups (Figure 1). However, no statistically significant difference in β-diversity was found in the ileum.

#### 3.2.2. Comparative Analyses of the Taxonomic Composition of the Ileum Microbial Community

Comparative analysis of the taxonomic composition at the genus level among the three groups was performed (Figure 2). In the 5-ASA-intolerant group, *Bacteroidota* were less abundant compared with the control group. On the contrary, *Firmicutes* were more abundant compared with the control group. *Lachnospiraceae*, *Oscillibacter*, and *Odoribacter* were less abundant in the 5-ASA-intolerant group than in the 5-ASA-tolerant group. In addition, *Methylobacterium* and *Brevundimonas* were more abundant in the 5-ASA-intolerant group than in the 5-ASA-tolerant group (Figure 3A–C and Figure 4A–E).

#### 3.2.3. Comparison between Two Groups of Ileal MAM (Figure 5)

*Lachnospiraceae ND3007* and *Monoglobus* were significantly more abundant in the control group, while *Ruminococcus gravus* and *Veillonella* were substantially more abundant in the 5-ASA-resistant group (Figure 5A). *Blautis*, *Fusicatenibacter*, *Ruminococcus torques*, *Ruminococcus*, *Butyrucicoccaceae,* and *Lachnociostridium* were significantly more abundant in the control group than in the 5-ASA intolerance group (Figure 5B). *Lachnospiraceae*, *Fusicatenibacter*, *Ruminococcus torques*, *Lachnospiraceae UCG-004*, *Dorea*, *Lachnoclostridium*, and *Ruminococcus gauvreauii* were significantly more abundant in the 5-ASA intolerance group than in the 5-ASA tolerance group (Figure 5C).

**Figure 5 biomedicines-12-02125-f005:**
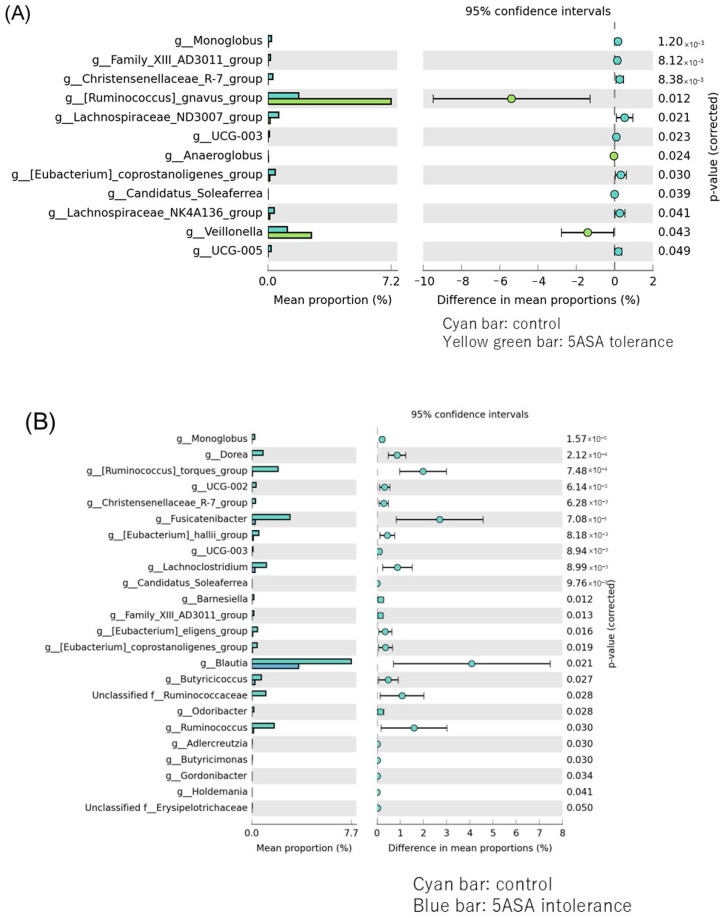
Comparison between two groups of ileal MAM. Bacteria species that showed significant differences between the two groups of (**A**) control (cyan bar) and 5-ASA tolerance (yellow green bar), (**B**) control and 5-ASA intolerance (blue bar), (**C**) 5-ASA intolerance and tolerance.

#### 3.2.4. The α-Diversity Indices among the Three Groups Using Sigmoid Colonic Mucus

The median indices of the observed features and Chao1r were significantly lower (*p* < 0.01) in the 5-ASA-intolerant group than in the controls. Moreover, significant differences in the observed features, Shannon entropy, and Chao1 indices were observed among the three groups (Figure 6). 

#### 3.2.5. Comparative Analyses of the Taxonomic Composition of the Microbial Community

Comparative analysis of the taxonomic composition at the genus level among the three groups was performed (Figure 7). *Proteobacteria* were less abundant in the 5-ASA-intolerant group than in HCs. *Actinobacteria* were more abundant in the 5-ASA-intolerant group than in HCs. *Oscillibacter* in the 5-ASA-intolerant group were more abundant than that in the 5-ASA-tolerant group (Figure 8A–C and Figure 9A,B).

#### 3.2.6. Comparison between Two Groups of Sigmoid Colonic MAM

*Ruminococcus gnavus* and *Incetrae sedis* were significantly more abundant in the 5-ASA tolerance group (purple group), while Prevotella, Sutterella, Leptotrichiam and Ruminococcaceae were substantially more abundant in the control group (red group) (Figure 10A). *Bifidobacterium* were significantly more abundant in the 5-ASA intolerance group, while Prevotella, Sutterella, Ruminococcaceae and Oribacterium were substantially more abundant in the control group (B). Anaerostipes and Lactobillus were significantly more abundant in the 5-ASA intolerance group, while *Incetae, Oscillibacter* and *Eubacterium nodatum* were substantially more abundant in the 5-ASA-resistant group (C).

## 4. Discussion

This study is the first to describe the relationship between 5-ASA intolerance UC and MAM. In addition, the result of MAM analysis using ileal and sigmoid colon samples was different in the 5-ASA-intolerant group. In particular, this was the first time that ileal samples were analyzed, and fewer bacteria, such as Oribacterium and Alistipes, in the 5-ASA-intolerant group were found to be effective for UC–fecal microbiota transplantation (FMT). Furthermore, fewer short fatty acid-producing bacteria, such as Lachnospiraceae and Ruminococcus, were found.

5-ASA intolerance development mechanisms are linked to an allergic reaction to 5-ASA preparations. Previous studies have revealed that the pathogenesis of UC involves many factors, such as immunological function, environmental factors, genetic predisposition, and intestinal microbiota, and suggested that the disturbance of the intestinal microbiota (dysbiosis) may be involved in 5-ASA intolerance [27,28].

Several studies have discussed beneficial butyrate-producing bacteria such as *Faecalibacterium prausnitzii* and Ruminococcaceae in 5-ASA-intolerant ulcerative colitis patients [39]. On the other hand, there are other reports of an increase in potentially harmful bacteria such as Enterobacteriaceae, Streptococcus, Veillonella, and Klebsiella in 5-ASA-intolerant UC patients [29]. These changes are in the feces, and our study used mucus. The present study examined changes in mucus and was characterized by a decrease in mucoprotective bacteria rather than an increase in harmful bacteria; the genera Ruminococcus and Lachnospiraceae were more abundant in healthy subjects and in UC long-term remission than in UC short-term remission and relapse patients. The abundance of *F. praegeri* was higher in healthy subjects and UC long-term remission than in UC short-term remission and relapse patients. *F. prausnitzii* and Lactobacillus were inversely proportional to UC disease severity, while Bacteroides, Hemophilus, and Veillonella were directly proportional [40].

In the 5-ASA-intolerant group, the abundance of Odoribacter and Alistipes, which are the key bacteria to effective FMT, was low in the ileal mucosa samples. An anaerobic Gram-negative bacterium, *Odoribacter splanchnicus*, generates SCFAs, including acetate, propionate, and butyric acid. SCFAs directly impact epithelial cells and influence Treg responses. The decrease in *O. splanchnicus* in IBD patients might stem from lower intestinal SCFA levels [41]. In a study by Lima et al., patients with UC who received FMT and had high levels of *O. splanchnicus* in their immunoglobulin A-coated microbiota had better clinical responses [42].

Alistipes may have defensive impacts against some illnesses, including liver fibrosis, colitis, cancer immunotherapy, and cardiovascular diseases [43]. *A. finegoldii* may be a protective species against colitis because *A. finegoldii* is diminished in mice with colitis. Thus, a study was performed in which microbiota-depleted mice were treated with oral DSS to induce colitis. The administration of *A. finegoldii* to mice resulted in a colitis severity comparable to that observed in wild-type mice [44]. In addition, previous studies have shown a common trend in Alistipes abundance in NOD2 knock-out murine microbiota profiles [45].

In fecal microbiota transplantation (FMT), fecal microbiota extracted from a healthy donor is placed into a patient’s intestine to rectify microbiome imbalance [46]. Several investigations, encompassing randomized controlled trials [47,48,49,50], systematic reviews, and meta-analyses [51,52,53], have indicated that fecal microbiota transplantation (FMT) is a viable treatment option for patients with active ulcerative colitis (UC). Evidence suggests that FMT may demonstrate greater efficacy compared to alternative microbial modulation strategies, such as the use of antibiotics, probiotics, and prebiotics.

Intestinal microbiota related to the gut–kidney axis were abundant in the sigmoid colon. Renal dysfunction is an adverse event of 5-ASA. The relationship between 5-ASA and gut bacteria is intricate, but it is not directly associated with kidney damage. 5-ASA can alter the gut microbiota composition, as shown in a study where it increased the relative abundance of Firmicutes and Actinobacteria, which include beneficial bacteria, and decreased the abundance of Bacteroidetes. This alteration in gut microbiota could contribute to various physiological effects, including impacts on the immune system. However, this study did not directly associate these changes with kidney damage. Alternatively, this study focused on the general effects of 5-ASA on gut bacteria and the potential implications for intestinal health and inflammation control. The detailed effects of these microbiota changes on kidney function or potential damage are not explicitly outlined in this study; thus, further research is necessary to establish any direct link.

The relationship between commonly used drugs and changes in the gut microbiome has been extensively studied, and several key findings are considered. One study showed that alimentary tract drugs such as proton pump inhibitors, osmotically acting laxatives, aminosalicylic acid, and similar agents can markedly alter the gut microbiome [54]. In this study, compared with the controls, the 5-ASA-intolerant group showed reduced ileal MAM diversity despite sampling in the absence of 5-ASA medication. Different results were observed when the administration of the current 5-ASA medication in the 5-ASA-intolerant group. Therefore, further research must be conducted to clarify the MAM of the 5-ASA-intolerant group during 5-ASA current medication.

This study also has some limitations. First, the number of cases studied is small for a single-center study. Second, fecal microbiota and metabolites have not been studied. Third, this study has a large number of inactive UC, and changes during the active phase are unknown. Moreover, the types of 5-ASA, pH-dependent, time-dependent, MMX, and SASP have not been studied. No 5-ASA intolerance symptom-specific studies have been conducted. Thus, further research must be conducted to clarify these limitations.

In this study, the relationship between 5-ASA intolerance and MAM was investigated. Interestingly, ileal MAM is different from sigmoid colon MAM in the 5-ASA-intolerant group. The present study showed changes in MAM and was characterized by a decrease in mucoprotective bacteria rather than an increase in harmful bacteria in 5-ASA intolerance patients.

## Figures and Tables

**Figure 1 biomedicines-12-02125-f001:**
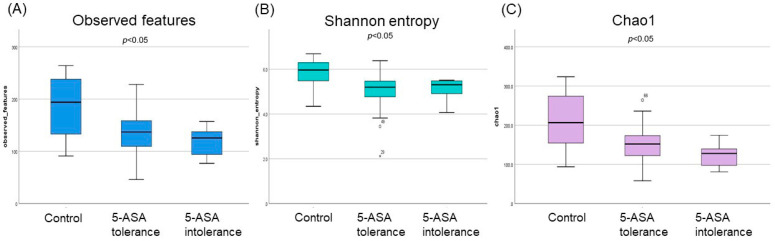
α-Diversity of ileal mucosa-associated microbiota (MAM); (**A**) observed features, (**B**) Shannon entropy, (**C**) Chao1.

**Figure 2 biomedicines-12-02125-f002:**
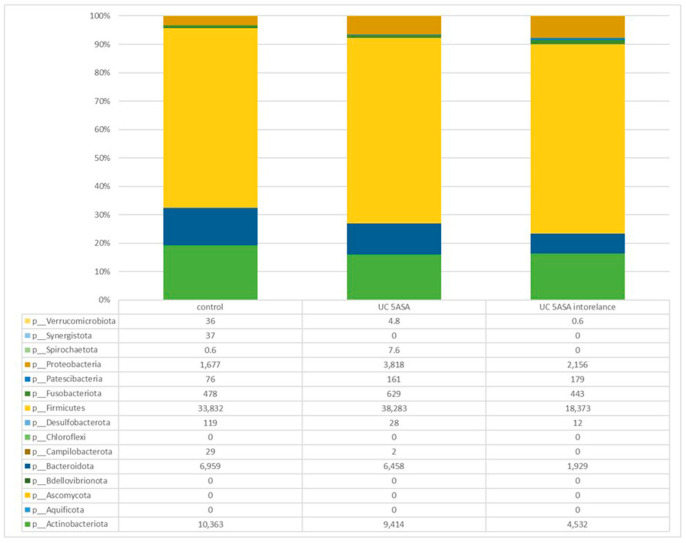
Taxonomic analysis of mucosa-associated microbiota (MAM) analysis using ileum mucus.

**Figure 3 biomedicines-12-02125-f003:**
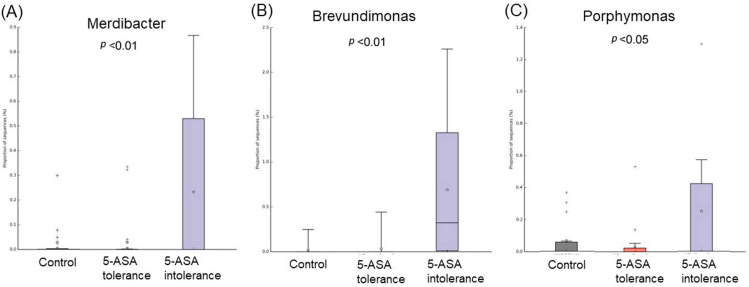
Comparison of three groups of ileal MAM: list of bacterial species that showed significant differences among three groups. The abundance of five intolerance groups was lower than the other two groups. (**A**) Merdibacter, (**B**) Brevundimonas, and (**C**) Porphymonas.

**Figure 4 biomedicines-12-02125-f004:**
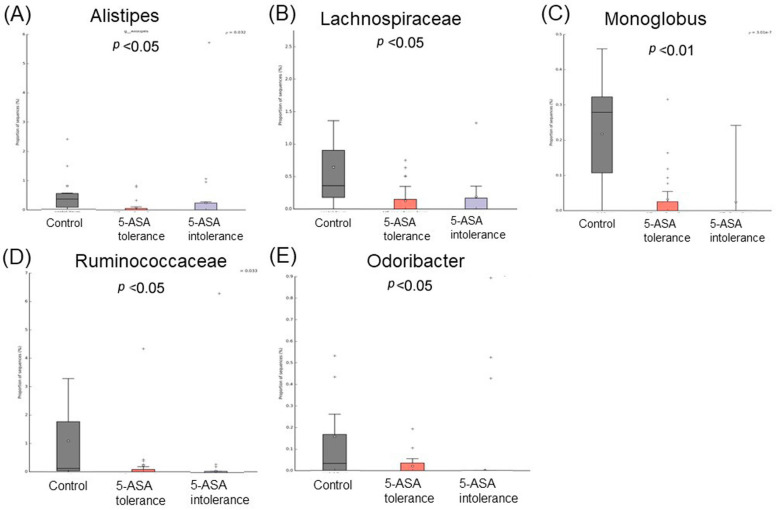
Comparison of three groups of ileal MAM: list of bacterial species that showed significant differences among three groups. The abundance of five intolerance groups was higher than the other two groups. (**A**) Alistipes, (**B**) Lachnospiraceae, (**C**) Monogiobus, (**D**) Ruminococcaceae, and (**E**) Odribacter.

**Figure 6 biomedicines-12-02125-f006:**
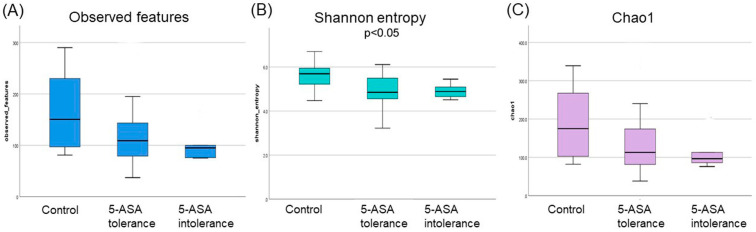
α-Diversity of colonic mucosa-associated microbiota (MAM): (**A**) observed features, (**B**) Shannon entropy, and (**C**) Chao 1.

**Figure 7 biomedicines-12-02125-f007:**
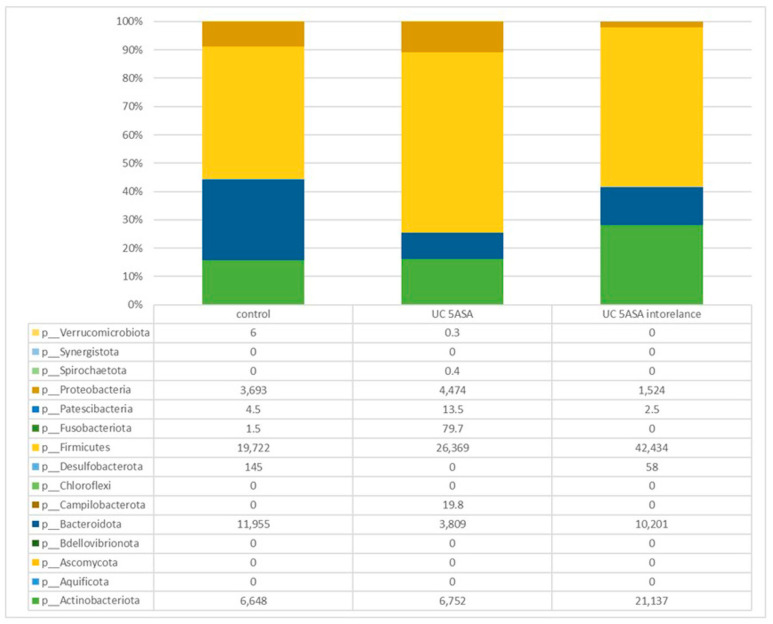
Taxonomic analysis of mucosa-associated microbiota (MAM) analysis using colonic mucus.

**Figure 8 biomedicines-12-02125-f008:**
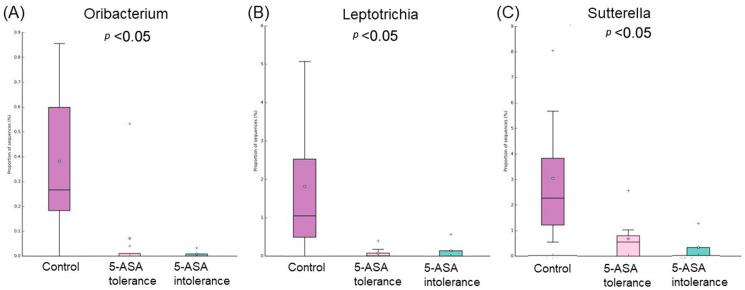
Comparison of three groups of colonic MAM: list of bacterial species that showed significant differences among three groups. The abundance of five intolerance groups was lower than the other two groups. (**A**) Oribacterium, (**B**) Laptotrichia, and (**C**) Sutterella.

**Figure 9 biomedicines-12-02125-f009:**
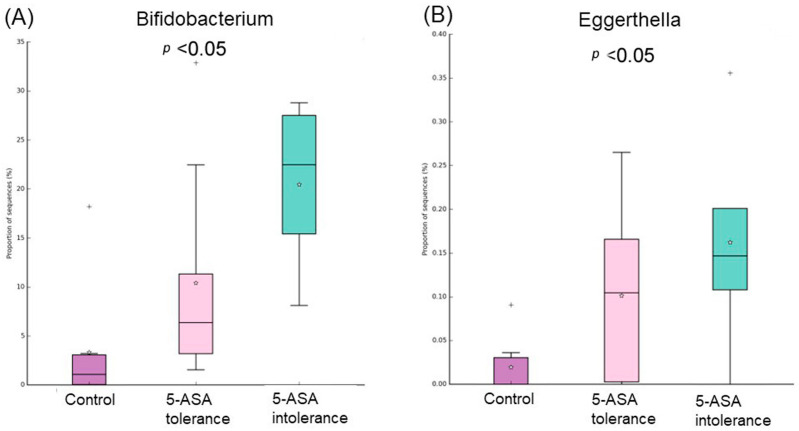
Comparison of three groups of colonic MAM: list of bacterial species that showed significant differences among three groups. The abundance of five intolerance groups was higher than the other two groups. (**A**) Bifidobacterium and (**B**) Eggerthella.

**Figure 10 biomedicines-12-02125-f010:**
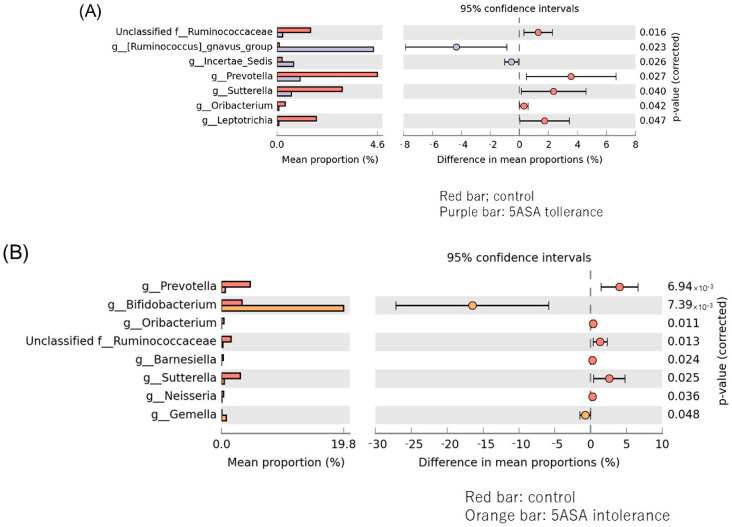
Comparison between two groups of sigmoid colonic MAM. Bacteria species that showed significant differences between the two groups of (**A**) control (red bar) and 5-ASA tolerance (purple bar), (**B**) control and 5-ASA intolerance (orange bar), (**C**) 5-ASA intolerance and tolerance.

**Table 1 biomedicines-12-02125-t001:** Patient background.

	Control	5-ASA Tolerance Group	5-ASA Intolerance Group	*p*
	19	38	15	
Age [mean ± SD]	56.2 ± 13.7	44.5 ± 16.1	48.1 ± 13.4	0.018
Gender (Male/Female)	7/12	16/22	6/9	NS
Disease duration (yr)	-	14.0 ± 9.1	14.1 ± 16.6	NS
Disease type	-	18/16/4	11/4/0	NS
Pan-colitis/Left side colitis/proctitis				
Type of 5ASA	-	7/11/18/2	4/3/5/3	NS
Time-dependant/pH-dependant/MMX/SASP				
Biologics	-	7(18.4%) [3/1/3/0]	11(100%) [1/0/1/9]	<0.01
TNFα/IL12-23/VED/JAK				
Endoscopic activity	-	29/7(95%)/2/0	7/2(82%)/2/0	NS
Mayo 0/1/2/3				
DLST	-	-	0/6/5	
Positive/negative/not inspected				

## Data Availability

The raw data supporting the conclusions of this article will be made available by the authors upon request. The data are restricted due to privacy reasons.

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
