# Peer review of "Characteristics of Mucosa-Associated Microbiota in Ulcerative Colitis Patients with 5-Aminosalicylic Acid Intolerance"

_biomedicines, 2024, doi:10.3390/biomedicines12092125_

Round 1

Reviewer 1 Report

Comments and Suggestions for Authors

The manuscript “Characteristics of mucosa-associated microbiota in Ulcerative colitis patients with 5-Aminosalicylic Acid Intolerance” is a well written paper that is well aligned with the journal title. The overall manuscript is good, there are some suggestions.

Abstract

The following sentence is somewhat ambiguous: “The association between 5-ASA intolerance and dysbiosis is investigated using fecal samples rather than brushing samples to analyze the mucosa-associated microbiota (MAM). Thus, this study aimed to investigate the MAM in patients with UC and compare those between 5-ASA tolerance and intolerance.”

It seems that the authors are trying to convey that previous studies have predominantly used fecal samples, whereas this study specifically utilized brushing samples to analyze the mucosa-associated microbiota (MAM).

Introduction

The sentence “However, there is no report about the association between 5-ASA intolerance and mucosa-associated microbiota, not fecal microbiota” is unclear. It is recommended to rephrase this for better clarity.

Could the authors please provide the date of the ethical approval letter? Additionally, it would be useful to include the study duration and specific dates.

The inclusion criteria section lacks clarity. It is important to elaborate on how the authors diagnosed Ulcerative Colitis in the participants. What diagnostic criteria or methods were used?

The discussion is not conclusive. There are several studies that have discussed that beneficial butyrate-producing bacteria like Faecalibacterium prausnitzii and Ruminococcaceae in 5-ASA intolerant UC patients. Moreover, 5-ASA intolerant UC patients show an increase in potentially harmful bacteria such as Enterobacteriaceae, Streptococcus, Veillonella, and Klebsiella

 https://academic.oup.com/ibdjournal/article/29/9/1362/7126988?login=false

https://link.springer.com/article/10.1007/s10238-022-00917-x

These studies need to be discussed

Comments on the Quality of English Language

The English quality is satisfactory; however, certain sentences need rephrasing to address grammatical and syntax mistakes.

Author Response

Thank you very much for the reviewers advice and comments. According to reviewer's comments, we rewrote and added new sentences, and new parts are in red-color, bold. We believe that the manuscript has been significantly improved by your comment. We hope the revised manuscript is suitable for publication in Biomedicine.

Reviewer 1

Comments and Suggestions for Authors

The manuscript “Characteristics of mucosa-associated microbiota in Ulcerative colitis patients with 5-Aminosalicylic Acid Intolerance” is a well written paper that is well aligned with the journal title. The overall manuscript is good, there are some suggestions.

Abstract

The following sentence is somewhat ambiguous: “The association between 5-ASA intolerance and dysbiosis is investigated using fecal samples rather than brushing samples to analyze the mucosa-associated microbiota (MAM). Thus, this study aimed to investigate the MAM in patients with UC and compare those between 5-ASA tolerance and intolerance.” It seems that the authors are trying to convey that previous studies have predominantly used fecal samples, whereas this study specifically utilized brushing samples to analyze the mucosa-associated microbiota (MAM).

Introduction

The sentence “However, there is no report about the association between 5-ASA intolerance and mucosa-associated microbiota, not fecal microbiota” is unclear. It is recommended to rephrase this for better clarity.

Response: Thank you very much for your kind comment. I rewrite this sentence in line14-16.

Could the authors please provide the date of the ethical approval letter? Additionally, it would be useful to include the study duration and specific dates.

Response: Thank you very much for kind advice. We added the new sentence in line 92-93; The Ethics Committee approval date is 2020-5-18, specific dates for study period 2020-6-29-2024-6-21.

The inclusion criteria section lacks clarity. It is important to elaborate on how the authors diagnosed Ulcerative Colitis in the participants. What diagnostic criteria or methods were used?

Response: Thank you very much for your advice. We add the new sentence about UC diagnosis in line 97-99.

The discussion is not conclusive. There are several studies that have discussed that beneficial butyrate-producing bacteria like Faecalibacterium prausnitzii and Ruminococcaceae in 5-ASA intolerant UC patients. Moreover, 5-ASA intolerant UC patients show an increase in potentially harmful bacteria such as Enterobacteriaceae, Streptococcus, Veillonella, and Klebsiella.

Response: Thank you very much for your advice. We add a new paragraph about results in line 259-271. And, we add a new sentence about the conclusion in line 318-320.

Reviewer 2 Report

Comments and Suggestions for Authors

In this study, Matsumoto et al. investigated the characteristics of mucosa-associated microbiota in ulcerative colitis (UC) patients with 5-aminosalicylic acid intolerance, revealing differences in microbial diversity and composition between tolerant and intolerant groups. While the topic of this study is meaningful and could add to our knowledge in this field, some issues within the manuscript need to be clarified. Here are some comments on this study:

1.       Line 27 “IBS”, IBS and IBD are two distinct gastrointestinal disorders.

2.       It is suggested that the names of phylum not be italicized.

3.       Line 47 “5-aminosalicylic acid (5-ASA)”, please define the abbreviation only when it first appears.

4.       Line 50 “UC.[8-10]”, citations need to be included the sentence (before the period).

5.       Line 66 “sugges”, please check the manuscript.

6.       Lines 76-80 “Mild adverse…”, these sentences don’t seem to be coherent and clear, please revise them.

7.       It is recommended that the authors state the significance of this study at the end of the introduction section.

8.       Line 141 “Greengenes database 141 (published May, 2013)”, the Greengene database is a fairly old version. It is recommended that a newer version be used.

9.       Section 2.6 “Alpha-diversity”, the method of beta diversity analysis is also included in this section.

10.     Line 146 “b-Diversity” should be beta diversity.

11.     Lines 148-150, please revise these sentences. PERMANOVA was performed by using the vegan package of R.

12.     Line 154 “paired one-way ANOVA”, how about the distribution and variance of the data?

13.     Ulcerative colitis (UC) is a remitting and relapsing inflammatory condition that affects the rectum and colon. Could the authors explain the reason for studying the ileal mucus microbiome?

14.     Significant difference labels are required in the figures.

15.     Line 189 “Figure 3A-H)(Figure 4A-C”, please check it.

16.     Lines 204-209 “(A). …(C)”, these sentences need to conform to the norms of scientific paper writing. The same issue is in section 3.2.6.

17.     Section “3.2.6. Comparison between two groups of sigmoid colonic MAM (Figure 10)”, please remove “(Figure 10)” in the title.

Author Response

Thank you very much for the reviewers advice and comments. According to reviewer's comments, we rewrote and added new sentences, and new parts are in red-color, bold. We believe that the manuscript has been significantly improved by your comment. We hope the revised manuscript is suitable for publication in Biomedicine.

Reviewer 2

Comments and Suggestions for Authors

In this study, Matsumoto et al. investigated the characteristics of mucosa-associated microbiota in ulcerative colitis (UC) patients with 5-aminosalicylic acid intolerance, revealing differences in microbial diversity and composition between tolerant and intolerant groups. While the topic of this study is meaningful and could add to our knowledge in this field, some issues within the manuscript need to be clarified. Here are some comments on this study:

  1. Line 27 “IBS”, IBS and IBD are two distinct gastrointestinal disorders.

Response: Thank you very much for your advice. I rewrite this word.

  1. It is suggested that the names of phylum not be italicized.

Response: Thank you very much for your advice. I rewrite this point.

  1. Line 47 “5-aminosalicylic acid (5-ASA)”, please define the abbreviation only when it first appears.

Response: Thank you very much for your advice. We change the abbreviation.

  1. Line 50 “UC.[8-10]”, citations need to be included the sentence (before the period).

Response: Thank you very much for your advice. I rewrite this point.

  1. Line 66 “sugges”, please check the manuscript.

Response: Thank you very much for your advice. I rewrite this point.

  1. Lines 76-80 “Mild adverse…”, these sentences don’t seem to be coherent and clear, please revise them.

Response: Thank you very much for your comments. I rewrite this sentence.

  1. It is recommended that the authors state the significance of this study at the end of the introduction section.

Response: Thank you very much for your comments. I added a new sentence about this study's aim.

  1. Line 141 “Greengenes database 141 (published May, 2013)”, the Greengene database is a fairly old version. It is recommended that a newer version be used.

Response: Thank you very much for your advice. I re-analyzed this study using Greengene version 13.8. Besides, I rewrite this section.

  1. Section 2.6 “Alpha-diversity”, the method of beta diversity analysis is also included in this section. Line 146 “b-Diversity” should be beta diversity.

Response: Thank you very much for your comments. We deleted the sentence about beta diversity analysis.

  1. Lines 148-150, please revise these sentences. PERMANOVA was performed by using the vegan package of R.

Response: Thank you very much for your advice. We rewrite the statical analysis including the sentence.

  1. Line 154 “paired one-way ANOVA”, how about the distribution and variance of the data?

Response: Thank you very much for your advice. I rewrite the sentence about the analysis.

  1. Ulcerative colitis (UC) is a remitting and relapsing inflammatory condition that affects the rectum and colon. Could the authors explain the reason for studying the ileal mucus microbiome?

Response: Thank your very much for your kind advice. We add the new sentence about why we study the ileal mucus microbiome in line 105-107.

  1. Significant difference labels are required in the figures.

Response: Thank you very much for your advice. We remake the new figures.

  1. Line 189 “Figure 3A-H)(Figure 4A-C”, please check it.

Response: Thank you for your comments. We rewrite these points.

  1. Lines 204-209 “(A). …(C)”, these sentences need to conform to the norms of scientific paper writing. The same issue is in section 3.2.6.

Response; Thank you for your kind comments. We rewrite these sentences.

  1. Section “3.2.6. Comparison between two groups of sigmoid colonic MAM (Figure 10)”, please remove “(Figure 10)” in the title.

Response: Thank you for your kind comments. We remove the word.

Round 2

Reviewer 1 Report

Comments and Suggestions for Authors

The manuscript is acceptable for publication now

Comments on the Quality of English Language

English is fine. Only minor issues

Author Response

Thank you very much for your comments. 

We check our English again.

Best regards,

Hiroshi Matsumoto

Reviewer 2 Report

Comments and Suggestions for Authors

I thank the authors for addressing all my comments

Author Response

Thank you very much for your advice.

We check our English again.

Best regards,

Hiroshi Matsumoto